# Testing the validity of a new scale designed to assess beliefs and perceptions about colorectal cancer and colorectal cancer screening in Malaysia: a principal component analysis

Tin Tin Su ,[1,2,3] Felix Oluyemi Adekunjo,[4] Desiree Schliemann ,[2] Christopher R Cardwell,[2] Mila Nu Nu Htay ,[3,5] Maznah Dahlui,[3,6] Siew Yim Loh,[7] Victoria L Champion,[8] Michael Donnelly[2]

For numbered affiliations see end of article.

**Correspondence to**
Professor Michael Donnelly; michael.donnelly@qub.ac.uk and
Professor Tin Tin Su; TinTin.Su@monash.edu

## ABSTRACT

**Objective** To conduct a cultural adaptation and validation of the Champion Health Belief Model Scale (CHBMS) for colorectal cancer (CRC) screening (CHBMS-CRC-M) in order to assess and investigate perceptions and beliefs about CRC screening in Malaysia.

**Designs and participants** The results from an evidence synthesis and the outcomes from an expert panel discussion were used to shape CHBMS scale content into an assessment of beliefs about CRC screening (CHBMS-CRC). This questionnaire assessment was translated into the official language of Malaysia. An initial study tested the face validity of the new scale or questionnaire with 30 men and women from various ethnic groups. Factorial or structural validity was investigated in a community sample of 954 multiethnic Malaysians.

**Setting** Selangor state, Malaysia.

**Results** The new scale was culturally acceptable to the three main ethnic groups in Malaysia and achieved good face validity. Cronbach's alpha coefficients ranged from 0.66 to 0.93, indicating moderate to good internal consistency. Items relating to perceived susceptibility to CRC 'loaded' on Factor 1 (with loadings scoring above 0.90); perceived benefits of CRC screening items loaded on factor 2 and were correlated strongly (loadings ranged between 0.63 and 0.83) and perceived barriers (PBA) to CRC screening (PBA) items loaded on factor 3 (range 0.30–0.72).

**Conclusion** The newly developed CHBMS-CRC-M fills an important gap by providing a robust scale with which to investigate and assess CRC screening beliefs and contribute to efforts to enhance CRC screening uptake and early detection of CRC in Malaysia and in other Malay-speaking communities in the region.

## STRENGTHS AND LIMITATIONS OF THIS STUDY

⇒ The development of the new Champion Health Belief Model Scale for colorectal cancer screening in Malaysia scale was informed by an evidence synthesis and iterative review by a multidisciplinary expert panel.

⇒ We followed the procedures set out in the relevant WHO guidelines for the process of translation and adaptation of instruments and the cross-cultural survey guidelines.

⇒ A large randomly selected sample from different ethnic groups was recruited.

⇒ Respondents were recruited from one state in Malaysia.

⇒ The questionnaire has not been tested for use with non-Malay-speaking ethnic groups.

## INTRODUCTION

The global burden of colorectal cancer (CRC) is increasing. Appropriate cancer control measures are needed to reduce increasing CRC mortality particularly in low-income and middle-income countries (LMICs) such as Malaysia where CRC is the second most common cancer.[1] According to GLOBOCAN report 2020, 6597 CRC cases were detected in 2020 (13.6% of all newly diagnosed cancer cases) and CRC-related deaths rose from 2565 to 3420 within 8 years.[2 3] It is estimated that about one-third to one-half of deaths due to CRC could be avoided through early presentation, detection and appropriate treatment.[4] However, over 50% of CRC patients in Malaysia present at late stages.[5] Research indicates that only 8% of people aged ≥50 years attend CRC screening, understanding about CRC screening tests is lacking[6 7] and willingness to participate in CRC screening is low.[8] According to Ho *et al*,[9] about half of cancer-related deaths in Malaysia could be avoided if cancer was detected and diagnosed early. Studies indicate that negative beliefs and perceptions may play a role in late presentation and diagnosis of advanced cancer among Malaysians.[10–12] Thus, there is

a need to develop a reliable and valid measurement tool to investigate perceptions and beliefs about screening.

The health belief model (HBM) provided the theoretical framework for this study.[7] Previous studies applied the model to breast cancer screening (BC) behaviour and, subsequently, the Champion Health Belief Model Scale (CHBMS-BC) was developed.[13] Further refinements and revisions led initially to the addition of a subscale—confidence regarding breast self-examination[14] and the addition of item questions to capture mammogram screening. The final revised CHBMS-BC for mammogram screening included three subscales: perceived susceptibility (PSU), benefits and barriers.[15] The CHBMS-BC has been translated into different languages and adapted and tested in different countries[16–19] including being used to assess the effectiveness of interventions which targeted women's participation in BC screening programme.[20–24] In addition, the CHBMS-BC has been adapted and validated to assess perceptions about BC screening among Malaysian women.[25 26]

As far as we are aware, a measure of the beliefs about CRC screening in Malaysia does not exist. The purpose of this study was to conduct a cultural adaptation of the CHBMS-BC questionnaire to develop a validated tool to investigate CRC screening beliefs in Malaysia.

## METHOD
### Evidence synthesis
We conducted a systematic review (according to PRISMA guidelines and a protocol was registered with PROSPERO) in order to synthesise the findings from Malaysia-based studies that investigated CRC symptoms awareness and barriers to cancer screening uptake.[27]

### Expert panel discussion
The CHBMS for mammogram screening was adapted for use in relation to CRC screening. An expert panel conducted the adaptation process based on the findings of the systemic review, differences in the nature of screenings and consideration of local scenarios. The panel of experts included three public health physicians, two cancer epidemiologists, one cancer advocate and one psychologist.

### Linguistic validation
The translation of the adapted scale (CHBMS-CRC-M) followed the procedures set out in the relevant WHO guidelines (WHO|Process of translation and adaptation of instruments, n.d.) and the Cross-Cultural Survey Guidelines (Adaptation—Cross-Cultural Survey Guidelines, n.d.). The questionnaire was translated into the local and official language of Malaysia (Bahasa Melayu). Two independent professional translators who are bilingual (English and Bahasa Melayu) conducted forward translation. We worked towards achieving a conceptual and culturally appropriate version rather than a literal translation. One of the researchers, who is a native

Malay speaker, compared both versions and resolved any discrepancies via discussions with the translators.

### Initial testing phase
We tested the face validity of the questionnaire with 30 (male/female) participants from the major ethnic groups in Malaysia (Malay, Chinese and Indian) in order to evaluate the extent to which the questionnaire was clear, simple and easy to understand, the presence of any inappropriate, redundant, incorrectly worded or missing items and the relevancy, flow and arrangement of the questionnaire. Trained interviewers undertook face-to-face interviews. Generally, the face validity study indicated that the newly developed CHBMS-CRC-M questionnaire was culturally acceptable to ethnic Malaysian men and women.

### Main study phase
The culturally validated CHBMS-CRC-M questionnaire was administered during the baseline data collection phase of the Be Cancer Alert Campaign.[28] The baseline survey occurred between January and March 2018 in Rawang subdistrict of the Gombak, Selangor State—the most populated state in Malaysia with a population above 5 million. Rawang subdistrict was chosen (in consultation with the Department of Statistics Malaysia: DOSM) because it contains a representative mix of ethnicities, age groups and income groups. The target sample comprised adult males and females who (1) lived in randomly selected households, (2) were above 40 years of age, (3) were fluent in Malay and (4) were able to answer independently without support from others. The age range was selected because populations in LMICs including Malaysia tend to be diagnosed with cancer at a younger age compared with high-income countries. The DOSM randomly selected 4368 houses from the 273 enumeration blocks (EBs—artificially created contiguous geographical areas with specific boundaries containing 100 households). We oversampled to take account of non-resident areas, business premises, non-respondents and survey-ineligible residents. The security guards and management offices in certain housing areas did not allow the survey to be conducted (this comprised a total of 656 houses across 36 EBs); 65 houses were non-residential business premises and 253 houses were unoccupied. The survey was conducted during official day time working hours due to our duty to ensure the safety of our interviewers. There did not appear to be residents at home in 1635 houses, most likely, because residents were working adults and were at their place of work when our interviewers called to these homes. A total of 1292 out of 1759 reachable households agreed to participate—a response rate of 73%. Finally, 954 participants who fulfilled the above-mentioned inclusion criteria completed the survey and the new CHBMS-CRC-M questionnaire. The percentage of missing data relating to items ranged from 0.4% to 2.7%.

Trained and supervised research assistants conducted face-to-face interviews after obtaining written informed consent from participants.

## Data analysis

SPSS V.22.0 statistical software was used to conduct descriptive statistics and a factor analysis (using a principal component analysis (PCA), varimax rotation, an eigenvalue of 1 and above and 0.3 as a cut-off point for loading onto a factor). Sampling adequacy was assessed using Kaiser-Meyer-Olkin (KMO), Bartlett's test of sphericity and a KMO value of 0.70 and above.[29 30] Reliability was assessed using Cronbach's alpha as a test of the internal consistency of items under each component of the scale: PSU to CRC screening, perceived benefits of CRC screening (PBE) and perceived barriers to CRC screening (PBA). We interpreted Cronbach Alpha scores to indicate as follows: 0.61–0.65—moderate, 0.71–0.91—good and 0.91–0.93—strong internal consistency in keeping with Taber.[31]

## Patient and public involvement

Key stakeholders and researchers with experience in cancer prevention and control were consulted at various stages including the cultural adaptation phase and during face validity testing of the new scale. The newly developed CHBMS-CRC-M will be posted on the research team's website (http://www.becanceralert.com/research/bcac/) and will be available freely for use by researchers and practitioners.

## RESULTS
## Evidence synthesis

Nine studies met review eligibility criteria but only three studies included data about barriers to CRC screening uptake. Each one of the three studies used a different self-developed questionnaire to assess barriers to screening and one study included data about domains related to the HBM in terms of PSU and benefits of screening.[6 32 33] 'Fear of result' from colorectal screening was the most commonly reported barrier in the three studies. Two studies reported the following comments in common which potentially could act as barriers or inhibiting factors: 'screening is embarrassing' (35% and 55%, respectively), 'did not have symptoms of CRC' (13% and 39%, respectively) and 'FOBT was not recommended by doctor' (11% and 35%, respectively).[30 31] Similarly, a further two studies from the review documented that 'fear of discomfort' during CRC screening (30% and 64%, respectively) and 'CRC screening is expensive' (23% and 53%, respectively).[6 33]

## Expert panel discussion

The expert panel used the three subscales (susceptibility, benefits and barriers) of the original scale to produce the 18-item CHBMS-CRC-M. A comparison of the item content of the original CHBMS and the newly adapted and developed CHBMS-CRC-M is presented in table 1. For example, the PBA subscale to CRC screening comprised 11 items which covered emotional, practical, service and financial barriers. The response format to every item was a 5-point Likert scale: (1) represented 'strongly disagree' while (5) represented strongly agree. Scores on each domain or subscale ranged as follows: susceptibility (3–15), benefits (4–20) and barriers (11–55). The explanatory note about CRC screening read: 'CRC screening is carried out to test whether or not someone has cancer. CRC can be detected through an examination of a stool and/or by examining a patient's colon and rectum with a camera that is inserted through their posterior or backside.'

The final data analysis comprised a sample of 954 survey participants. Table 2 presents descriptive statistics for the sample. For example, 62% were male, 82% were married, around a quarter had at least above certificate education level; mean age was 50.84 years (SD=16.85) and, on average, there were five residents per household (SD=2.28).

## Principal component analysis

The sample size (n=954) in relation to the new 18-item scale and 5-point Likert response was more than required in order to conduct a robust PCA.[34] A PCA was conducted on the 18 items of the CHBMS with varimax rotation. We applied a PCA to obtain the component or factor solution based on the linear combination of observed items.[35]

Varimax rotation was used to clarify the relationships between components and items.[36] Researchers applied PCA and varimax rotation during the validation of the original CHBMS for BC screening,[20] and the translated Malay version of CHBMS-BC-M.[37] We followed a similar analytic process to the process that was used in the previously published articles (referenced above) in keeping with the original theoretically informed investigative approach to BC and mammogram screening. A key aim of the analysis was to test the appropriateness and acceptability of the adapted and supplemented item content in the context of a very different type of cancer from BC; and to conduct a PCA to investigate the nature and degree to which the item content grouped together to represent or explain beliefs about CRC and potential use of CRC screening. Ideally, further field testing would occur and at some later stage a confirmatory factor analysis might be undertaken.

The KMO test verified sampling adequacy for the conduct of a PCA (KMO=0.73) and Bartlett's test of sphericity ($\chi^2$=5093.73) confirmed that the sample size was adequate. The analysis obtained eigenvalues for the factors each of which exceeded Kaiser's criterion of 1 and in combination explained 44% of variance (table 3).

Items relating to PSU to CRC 'loaded' on factor 1 (with loadings above 0.90); PBE of CRC screening items loaded on factor 2 and were correlated strongly (loadings ranged between 0.63 and 0.83); and PBA to CRC screening items loaded on factor 3 (range 0.30–0.72). Table 4 presents

**Table 1** Comparison of items from the original CHBMS for mammogram screening and newly developed CHBMS-CRC-M

| Original CHBMS for mammogram screening | CHBMS_CRC-M |
|---|---|
| Susceptibility | |
| 1. It is likely that I will get breast cancer. | 1. It is likely that I will get colorectal cancer. |
| 2. My chances of getting breast cancer in the next few years are great. | 2. My chances of getting colorectal cancer in the next few years are high. |
| 3. I feel I will get breast cancer sometime during my life. | 3. I feel I will get colorectal cancer sometime during my life. |
| Benefits | |
| 1. If I get a mammogram and nothing is found, I do not worry as much about breast cancer. | 1. If I get colorectal cancer screening and nothing is found, I will worry less about colorectal cancer. |
| 2. Having a mammogram will help me find breast lumps early. | 2. Having colorectal cancer screening will help me find colorectal cancer early. |
| 3. If I find a lump through a mammogram, my treatment for breast cancer may not be as bad. | 3. If I find abnormalities through colorectal cancer screening my treatment for colorectal cancer may not be as bad. |
| 4. Having a mammogram is the best way for me to find a very small lump. | |
| 5. Having a mammogram will decrease my chances of dying from breast cancer. | 4. Having colorectal cancer screening will decrease my chances of dying from colorectal cancer. |
| Barriers | |
| 1. I am afraid to have a mammogram because I might find out something is wrong. | 1. I am afraid to have colorectal cancer screening because I might find out something is wrong. |
| 2. I am afraid to have a mammogram because I don't understand what will be done. | 2. I am afraid to have colorectal cancer screening because I don't understand what will be done. |
| 3. I don't know how to go about getting a mammogram. | 3. I don't know how to go about getting colorectal cancer screening. |
| 4. Having a mammogram is too embarrassing. | 4. Having colorectal cancer screening is too embarrassing. |
| 5. Having a mammogram takes too much time. | 5. Having colorectal cancer screening takes too much time. |
| 6. Having a mammogram is too painful. | 6. Having colorectal cancer screening is too painful. |
| 7. People doing mammograms are rude to women. | 7. People doing colorectal cancer screening are rude to patients. |
| 8. Having a mammogram exposes me to unnecessary radiation. | |
| 9. I cannot remember to schedule a mammogram. | |
| 10. I have other problems more important than getting a mammogram. | 8. I have other problems more important than getting colorectal cancer screening. |
| 11. I am too old to need a routine mammogram. | 9. I am not the right age to need a routine colorectal cancer screening. |
| | 10. I cannot afford to get colorectal cancer screening. |
| | 11. I don't have the encouragement I need from my close relatives to attend colorectal cancer screening. |

CHBMS-CRC-M, Champion Health Belief Model Scale for colorectal cancer screening in Malaysia.

the rotated factor matrix and three-factor solution (items greater than 0.3 were retained as recommended by best practice).[30]

### Internal reliability or consistency
The PSU to CRC indicated high reliability (Cronbach's α=0.93, with all items strongly correlated around 0.80); the PBE of CRC screening (PBE) had fairly high overall internal consistency (Cronbach's α=0.78 though inter-item correlations ranged from 0.31 to 0.57); and the reliability test for PBA to CRC screening was moderate

(Cronbach's α=0.66 with some inter-item correlations below 0.30).

### DISCUSSION
Although annual CRC screening is recommended for people aged ≥50 years in Malaysia,[38] a population-based screening programme is not in place and efforts so far to improve uptake have been selective and sporadic.[39] The absence of a population-based screening programme and the development of advanced cancer impacts

**Table 2** Sociodemographic characteristics (n=954)

| Characteristic | No | Percentage |
|---|---|---|
| Gender | | |
| Male | 361 | 38 |
| Female | 593 | 62 |
| Nationality | | |
| Malaysian | 871 | 91 |
| Non-Malaysian | 80 | 8 |
| Marital status | | |
| Single | 31 | 3 |
| Married | 783 | 82 |
| Divorcee and widow | 136 | 15 |
| Religion | | |
| Islam | 585 | 61 |
| Christianity | 35 | 4 |
| Buddhism | 95 | 10 |
| Hinduism | 226 | 24 |
| Others | 11 | 1.1 |
| Ethnicity | | |
| Malay | 516 | 54 |
| Chinese | 110 | 12 |
| Indian | 264 | 28 |
| Others | 64 | 6.7 |
| Education | | |
| No formal education | 73 | 8 |
| Never completed primary school | 79 | 8 |
| Completed primary school | 190 | 20 |
| Completed secondary school | 386 | 41 |
| Certificate A-Level/STPM/HSC | 99 | 10 |
| Tertiary education | 125 | 13 |
| Current job | | |
| Civil servant | 26 | 3 |
| Private sector employee | 194 | 20 |
| Self-employed | 128 | 13 |
| Retiree | 116 | 12 |
| Homemakers | 377 | 40 |
| Unemployed | 113 | 12 |
| Main occupation | | |
| Manager | 13 | 1 |
| Professionals | 50 | 5 |
| Technicians and associate professionals | 31 | 3 |
| Clerical support workers | 20 | 2 |
| Service and sales workers | 110 | 12 |
| Skilled agricultural, forestry and fishery workers | 6 | 0.6 |

**Table 2** Continued

| Characteristic | No | Percentage |
|---|---|---|
| Craft and related trades workers | 5 | 0.5 |
| Plant and machine-operators and assemblers | 19 | 2 |
| Elementary occupations | 87 | 9 |
| Healthcare professional (ie, doctors, nurses, etc) | 5 | 0.5 |
| Monthly family income | | |
| Below RM2000 | 342 | 36 |
| RM2000–RM3000 | 201 | 21 |
| RM3000–RM4000 | 118 | 12 |
| Above RM4000 | 147 | 16 |

negatively on survival,[5 40] requires more intensive treatment, increases healthcare resource utilisation and places additional financial burden on households. The healthcare system and context in Malaysia requires systematic research attention in order to ensure the successful implementation of CRC screening, particularly in terms of the promotion, acceptance of and participation in, CRC screening among the general population. Misperception of risk of cancer, denial of the existence of the disease, emotional barriers and negative beliefs about cancer screening generally affect screening uptake and lead to delay in seeking medical help.

An in-depth understanding about how adults perceive their susceptibility to CRC, the benefits of attending screening and barriers to CRC screening is crucial for the successful implementation of CRC screening and enhancement of screening coverage. A systematic review comprising 30 articles evaluated the association between HBM domains and CRC screening in the community.[41] The review found that higher scores for the domains, PSU and benefits, and lower scores for the barriers domain were associated with positive screening behaviour; and cues to action were associated with CRC screening adherence.[41] The results of the review provide support for the potential benefits of using the new scale in terms of improved understanding and providing insights into the design of screening uptake interventions.

We adapted, translated and tested the CHBMS-CRC-M in order to create a means of evaluating Malaysians' beliefs or perceptions about CRC screening. The cultural adaptation of the CHBMS-CRC-M was grounded robustly via an evidence synthesis, an expert panel discussion and linguistic validation exercise. The scale was proven to have good face validity in a relatively small separate survey from the main study which assessed its structural validity in a large randomly selected community sample of multiethnic Malay-speaking men and women.

The results of our validation study showed that the newly developed CHBMS-CRC-M is a reliable and valid instrument for measuring beliefs regarding CRC screening

**Table 3** Total variance explained

| Item | Initial eigenvalues | | | Extraction sums of squared loadings | | | Rotation sums of squared loadings | | |
|---|---|---|---|---|---|---|---|---|---|
| | Total | % of variance | Cumulative % | Total | % of variance | Cumulative % | Total | % of variance | Cumulative % |
| 1 | 2.966 | 16.476 | 16.476 | 2.966 | 16.476 | 16.476 | 2.685 | 14.914 | 14.914 |
| 2 | 2.682 | 14.898 | 31.374 | 2.682 | 14.898 | 31.374 | 2.672 | 14.842 | 29.756 |
| 3 | 2.182 | 12.123 | 43.497 | 2.182 | 12.123 | 43.497 | 2.473 | 13.740 | 43.497 |
| 4 | 1.393 | 7.738 | 51.234 | | | | | | |
| 5 | 1.089 | 6.048 | 57.282 | | | | | | |
| 6 | 1.046 | 5.810 | 63.092 | | | | | | |
| 7 | 0.923 | 5.130 | 68.222 | | | | | | |
| 8 | 0.874 | 4.856 | 73.078 | | | | | | |
| 9 | 0.777 | 4.317 | 77.395 | | | | | | |
| 10 | 0.731 | 4.059 | 81.453 | | | | | | |
| 11 | 0.643 | 3.573 | 85.027 | | | | | | |
| 12 | 0.632 | 3.514 | 88.540 | | | | | | |
| 13 | 0.569 | 3.159 | 91.700 | | | | | | |
| 14 | 0.474 | 2.631 | 94.331 | | | | | | |
| 15 | 0.380 | 2.112 | 96.443 | | | | | | |
| 16 | 0.278 | 1.547 | 97.990 | | | | | | |
| 17 | 0.250 | 1.388 | 99.379 | | | | | | |
| 18 | 0.112 | 0.621 | 100.000 | | | | | | |

among Malay-speaking multiethnic Malaysians. Cronbach's alpha coefficients for the three subscales ranged from 0.66 to 0.93, which indicated moderate to high internal consistency. The PCA revealed that each item of the scale showed acceptable item correlations (greater than the criterion of 0.30). The relevant items from the 18-item pool loaded separately and sufficiently onto three factors or components in a pattern and manner that was consistent with the three subscales of PSU to CRC, PBE of CRC screening and PBA to CRC screening. Barriers tend to vary in nature (eg, emotional, practical, service and financial barriers) and a given barrier may have more, less or the same perceived salience for different respondents depending on a range of psychosocial health and life experiences (with resultant implications for intervention design and delivery). Arguably, given this point, it is surprising that a Cronbach's alpha for PBA domain of 0.66 was achieved although less than the convention of <0.7. We interpreted 0.6 as indicating an acceptable moderate level of internal reliability (in keeping with other researchers, eg, Taber, 2018) and viewed the PBA domain to be a single discrete domain that was consistent with the HBM.[20]

The new measure appears to have good structural and construct validity for the assessment of beliefs about CRC and CRC screening as well as good face validity and internal reliability. Our findings are in broad agreement with a study in Korea that translated and validated the CHBMS for CRC screening.[42] However, the analysis of the much longer 45-item Korean version generated an

additional domain or component (in addition to the above three components) that assessed beliefs related to personal self-efficacy. In contrast, our study led to a shorter version of CHBMS that was adapted, translated and validated to be applicable in the Malay-speaking community. We recognise the long-established importance of self-efficacious beliefs to performing positive health behaviours and avoiding harmful behaviours and, so, researchers may wish to avail of the option of adding one of the many existing short, easy to use well-validated self-efficacy scales for cancer screening though there is a need to be mindful of respondent burden and keeping respondents engaged sufficiently to achieve the goal of CRC screening uptake.

The newly developed CHBMS-CRC-M fills an important gap by providing a robust scale with which to investigate and assess CRC screening beliefs and, so, contribute to the development and implementation of culturally sensitive and effective programmes that will enhance CRC screening uptake and early detection of CRC in Malaysia. This scale is suitable, too, for use with other Malay-speaking communities in the region such as Indonesia, Singapore, Philippine, southern Thailand and southern Myanmar.

### Strengths and limitations
This study applied a thorough research-informed validation process to develop the new CHBMS-CRC-M scale. The reasonably large and randomly selected sample from the different ethnic groups was recruited to ascertain

| Subscales and items | Factor loadings | Factor loadings | Factor loadings |
|---|---|---|---|
| Susceptibility | | | |
| PSU1 | 0.944 | | |
| PSU2 | 0.943 | | |
| PSU3 | 0.903 | | |
| Benefits | | | |
| PBE1 | | 0.627 | |
| PBE2 | | 0.826 | |
| PBE3 | | 0.799 | |
| PBE4 | | 0.760 | |
| Barriers | | | |
| PBA1 | | | 0.716 |
| PBA2 | | | 0.702 |
| PBA3 | | | 0.305 |
| PBA4 | | | 0.619 |
| PBA5 | | | 0.391 |
| PBA6 | | | 0.429 |
| PBA7 | | | 0.428 |
| PBA8 | | | 0.533 |
| PBA9 | | | 0.419 |
| PBA10 | | | 0.304 |
| PBA11 | | | 0.321 |

**Table 4** CHBMS-CRC-M item loadings for each factor

CHBMS-CRC-M, Champion Health Belief Model Scale for colorectal cancer screening in Malaysia; PBA, perceived barriers; PBE, perceived benefits; PSU, perceived susceptibility.

cultural acceptability and applicability across the multi-ethnic Malay-speaking people in Malaysia. However, it is important to note that respondents were recruited from Selangor State in Malaysia, and the scale/questionnaire has not been tested with non-Mala-speaking ethnic groups.

## CONCLUSION

There is an urgent need to improve CRC screening uptake in Malaysia and efforts are needed to address negative beliefs and barriers to CRC screening. The newly developed CHBMS-CRC-M is a reliable and valid scale to measure the beliefs on CRC screening in Malay-speaking communities in Malaysia. In addition to using the scale in public health research and service development, the information gathered by using CHBMS-CRC-M may prove to be useful for primary care physicians and staff in terms of helping to identify and reduce misperceptions and barriers to cancer screening, thereby facilitating attendance and improving uptake.

**Author affiliations**
[1]South East Asia Community Observatory (SEACO), Jeffery Cheah School of Medicine and Health Sciences, Monash University Malaysia, Bandar Sunway, Selangor, Malaysia
[2]Centre for Public Health and UKCRC Centre of Excellence for Public Health, Queen's University Belfast, Belfast, UK
[3]Centre for Pooulation Health (CePH), Department of Social and Preventive Medicine, Faculty of Medicine, University of Malaya, Kuala Lumpur, Malaysia
[4]Department of Economics, Faculty of Social Sciences, Lagos State University, Lagos, Nigeria
[5]Department of Community Medicine, Faculty of Medicine, Manipal University College Malaysia, Melaka, Malaysia
[6]Department of Research Development and Innovation, University Malaya Medical Centre, Kuala Lumpur, Malaysia
[7]Department of Rehabilitation Medicine, Faculty of Medicine, University of Malaya, Kuala Lumpur, Malaysia
[8]School of Nursing, Indiana University, Bloomington, Indiana, USA

**Acknowledgements** The authors would like to acknowledge the support of Dr Nor Saleha Binti Ibrahim Tamin and Dr Saunthari Somasundaram for the expert panel discussion and Ms Darishiani Paramasivam for systematic review and field data collection.

**Collaborators** not applicable.

**Contributors** MDo and TTS conceptualised and planned the project with consultation with VLC. TTS, MDa, SYL and DS conducted expert panel discussion and cultural adaptation process. FOA conducted the analysis. FOA, CLC, MNNH and TTS interpreted the results. TTS drafted the manuscript with MDo and FOA. All authors reviewed and approved the final manuscript. MDo is responsible for the overall content as guarantor.

**Funding** This research was supported by the UK Medical Research Council—NewtonUngku Omar Funding (MR/P013910/1) and the MRC UKRI GCRF (MR/S014349/1).

**Disclaimer** The funder had no role in the design of the study, collection, analysis, and interpretation of data or writing the manuscript.

**Competing interests** None declared.

**Patient and public involvement** Patients and/or the public were involved in the design, or conduct, or reporting, or dissemination plans of this research. Refer to the Methods section for further details.

**Patient consent for publication** Consent obtained directly from patient(s).

**Ethics approval** This study involves human participants and was approved by the Medical Research Ethics Committee, University Malaya Medical Centre (ID: 2016126-4668) and by the National Medical Research Register (ID: NMRR-17-2788-35613). Participants gave informed consent to participate in the study before taking part.

**Provenance and peer review** Not commissioned; externally peer reviewed.

**Data availability statement** Data are available on reasonable request. Data are available from the corresponding authors, MDo and TTS, on reasonable request.

**ORCID iDs**
Tin Tin Su http://orcid.org/0000-0003-0387-6406
Desiree Schliemann http://orcid.org/0000-0002-8746-3002
Mila Nu Nu Htay http://orcid.org/0000-0003-2506-3473

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
