## [Reviewer comments · BMJ Open]

ARTICLE DETAILS

TITLE (PROVISIONAL)	Testing the validity of a new scale designed to assess beliefs and perceptions about colorectal cancer and colorectal cancer screening in Malaysia: A principal component analysis
AUTHORS	Su, Tin; Adekunjo, Felix; Schliemann, Desiree; Cardwell, Christopher; Htay, Mila; Dahlui, Maznah; Loh, Siew_Yim; Champion, Victoria; Donnelly, Michael

VERSION 1 – REVIEW

REVIEWER	Wan Sulaiman, Wan Aliaa Universiti Putra Malaysia, Neurology
REVIEW RETURNED	24-Feb-2021

GENERAL COMMENTS	A well written manuscript with scientifically sound methodology. Discussion are thorough with critical appraisal of literature. Accept for publication.
---

REVIEWER	De Guzman, Roselle Manila Central University-Filemon D Tanchoco Medical Foundation Hospital
REVIEW RETURNED	30-Jun-2021

GENERAL COMMENTS	This paper discusses the validity of the Champion's Health Belief Model Scale for colorectal cancer in Malaysia. This is the first that can be potentially of use for the country. The main contribution of this paper is that it addresses the issues of colorectal cancer control through screening and early diagnosis. The authors did a meticulous validation process and randomized a fairly large sample of the study population from the Selangor state. Pre-existing research studies are appropriately referenced. The methods are sufficiently detailed that include description of how subjects were randomly selected. The results are clearly presented. The tables stand on their own and are properly referred to in the discussion of the results. Major comments: 1. There was informed consent from patients but no IRB approval was mentioned.2. Methods. There were areas with 656 houses that survey was not conducted. There were also 1635 unreachable houses. These are significant numbers. Could these be further clarified?3. Some of the recent papers on the topic are not cited, among these are the works published by Lau (2020, Singapore) and Lee (2020, Korea). It would be interesting to see some discussion on the findings of these papers. Minor comments:
--

	1. Globocan 2020 is the newest cancer data. This can be used as one of the references supporting the background information instead of the Globocan 2012.
--	---

REVIEWER	Gasper, Harry Toowoomba and Darling Downs Health Service District, Cancer Care
REVIEW RETURNED	08-Jul-2021

GENERAL COMMENTS	Many thanks for the opportunity to review the manuscript for your study, which I enjoyed reading. This is a validation study, looking into the development of a measurement tool to assess perceptions and beliefs about screening in the Malay population. The investigators have adapted the Champion Health Belief Model Scale for use in colorectal cancer and in turn sought to translate it into a questionnaire that would be conceptually and culturally appropriate to a Malay population. This was subsequently tested and validated, with results reported. The study is well described and clearly written up in a flowing and easy to follow manuscript. The methods are clearly described and repeatable, and appear suitable for the task in hand, although I would be happy if a trained statistician look over the paper for further rigour as to the appropriateness of the tests news. The scale itself appears appropriate to investigate the beliefs around screening, it is not cumbersome, but is sufficiently detailed and covers three main domains. One small, but important point was that I could not find a clear statement in the write up regarding the ethics oversight for the study. The results of the study are clearly documented in sections. The sample size and response rate was adequate, and the tables relate clearly to the text. The conclusions made are not exaggerated, and are supported by the presented data. The discussion linked in this study to the objectives and included a section on strengths and limitations. The newly translated scale seems to be a robust tool for the purpose it has been designed for. I would be interested to know more about how the answers from the questionnaire will be used to help plan a screening program. What numbers of uptake are needed in order to make the colorectal cancer screening work and what interventions will help to improve this? If this information is to be part of a later write up, I would be most interested to read it. To summarise, this is a neatly executed and well presented study looking to validate a tool to assess beliefs and perceptions of colorectal cancer screening in a Malay population. I support publication of this study, and look forward to seeing how it may be used to improve understand the reported screening rates in the under 50's.
---

REVIEWER	Sidani, S Ryerson University
REVIEW RETURNED	17-Nov-2021

GENERAL COMMENTS	Overall, the authors provided limited explanations of the analytic decisions made.
--

	The descriptive analyses at the item and factor of component levels are appropriate; however, these findings were not reported. In particular, the percentage of missing data on each item was not reported, yet this has implications in interpreting the findings. Factor analysis may be appropriate, although a confirmatory factor analysis could have been more useful in validating the proposed factorial structure of the measure, which was derived from theory (health beliefs model). The rationale for selecting: 1) principle component analysis instead of principal axis factoring (commonly used when factors represent concepts underlying responses to items), and 2) varimax rotation instead of oblique rotation (commonly used when the underlying factors are expected, in real world, to be correlated), is not presented. Please, explain the decisions. The cut-off for eigenvalue is usually > 1.00 (not 0.3 as implied in the respective sentence) and for factor loading > 0.3 (though some use > 0.5). Please revise the sentence accordingly. Also, was a difference of 0.2 in the loadings of the same item on 2 factors used as a criterion to determine which factor the item loads on? Please, explain. The Cronbach's alpha coefficient is set at > 0.70 for newly developed measure. What is the reason for setting it at 0.60 in this study? The low (< 0.70) alpha coefficient value for the barriers subscale (or component) is attributable to the content of its items. The items assess different emotional, practical, service, and financial barriers. Participants may perceive these barriers differently. This also has implications for the interpretation of the findings and for the use of the measure in practice; in practice, it would be more meaningful to determine the specific type of barrier perceived by patients, as this inform the selection of most relevant intervention.
--	--

VERSION 1 – AUTHOR RESPONSE

Reviewer: 1

Dr. Wan Sulaiman, Universiti Putra Malaysia

Comments to the Author:

A well written manuscript with scientifically sound methodology. Discussion are thorough with critical appraisal of literature. Accept for publication.

Response: Thank you for the positive consideration of our manuscript and recommendation for publication.

Reviewer: 2

Dr. Roselle De Guzman, Manila Central University-Filemon D Tanchoco Medical Foundation Hospital

Comments to the Author:

This paper discusses the validity of the Champion's Health Belief Model Scale for colorectal cancer in Malaysia. This is the first that can be potentially of use for the country. The main contribution of this paper is that it addresses the issues of colorectal cancer control through screening and early diagnosis. The authors did a meticulous validation process and randomized a fairly large sample of the study population from the Selangor state. Pre-existing research studies are appropriately referenced. The methods are sufficiently detailed that include description of how subjects were randomly selected. The results are clearly presented. The tables stand on their own and are properly referred to in the discussion of the results.

Response: Thank you for the positive feedback and insightful comments on our manuscript.

Major comments:

1. There was informed consent from patients but no IRB approval was mentioned.

Response: Detailed information on ethic committees and approval numbers were added in the method section

2. Methods. There were areas with 656 houses that survey was not conducted. There were also 1635 unreachable houses. These are significant numbers. Could these be further clarified?

Response: The reasons why it was not possible to assess a resident in 656 houses and contact a member of 1635 households were as follows:

The security guards and housing management offices for 656 houses from 36 EBs did not grant permission to the research team to conduct the survey.

The survey was conducted during official daytime working hours due to our duty to ensure the safety of our interviewers. There did not appear to be residents in 1,635 houses, most likely, because residents were working adults and were at their place of work when our interviewers called to these homes.

3. Some of the recent papers on the topic are not cited, among these are the works published by Lau (2020, Singapore) and Lee (2020, Korea). It would be interesting to see some discussion on the findings of these papers.

Response: We included recent publications on HBM and CRC screening by Lau (2020, Singapore) and Lee (2019, Korea) in the discussion.

Minor comments:

1. Globocan 2020 is the newest cancer data. This can be used as one of the references supporting the background information instead of the Globocan 2012.

Response: We updated information based on GLOBOCAN 2020

Reviewer: 3

Dr. Harry Gasper, Royal Brisbane and Women's Hospital, The University of Queensland

Comments to the Author:

Many thanks for the opportunity to review the manuscript for your study, which I enjoyed reading.

This is a validation study, looking into the development of a measurement tool to assess perceptions and beliefs about screening in the Malay population. The investigators have adapted the Champion Health Belief Model Scale for use in colorectal cancer and in turn sought to translate it into a questionnaire that would be conceptually and culturally appropriate to a Malay population. This was subsequently tested and validated, with results reported.

The study is well described and clearly written up in a flowing and easy to follow manuscript. The methods are clearly described and repeatable, and appear suitable for the task in hand, although I would be happy if a trained statistician look over the paper for further rigour as to the appropriateness of the tests news. The scale itself appears appropriate to investigate the beliefs around screening, it is not cumbersome, but is sufficiently detailed and covers three main domains. The results of the study are clearly documented in sections. The sample size and response rate was adequate, and the tables relate clearly to the text. The discussion linked in this study to the objectives and included a section on strengths and limitations. The newly translated scale seems to be a robust tool for the purpose it has been designed for. To summarise, this is a neatly executed and well presented study looking to validate a tool to assess beliefs and perceptions of colorectal cancer screening in a Malay population. I support publication of this study, and look forward to seeing how it may be used to improve understand the reported screening rates in the under 50's.

Response: Thank you for the positive and insightful comments, and support publication of this study

One small, but important point was that I could not find a clear statement in the write up regarding the ethics oversight for the study.

Response: Detailed information on ethic committees and approval numbers were added in the method section

The conclusions made are not exaggerated, and are supported by the presented data.

Response: We revised the conclusion.

I would be interested to know more about how the answers from the questionnaire will be used to help plan a screening program. What numbers of uptake are needed in order to make the colorectal cancer screening work and what interventions will help to improve this? If this information is to be part of a later write up, I would be most interested to read it.

Response: Thank you for your interest. Information gathered from this newly developed CHBM-CRC-M will be published in a separate paper after this validation paper is accepted. We further developed CRC screening intervention protocol based on the finding of this survey and the manuscript is under review.

We also strengthened the discussion of the manuscript.

Reviewer: 4

Dr. S Sidani, Ryerson University

Comments to the Author:

Overall, the authors provided limited explanations of the analytic decisions made.

The descriptive analyses at the item and factor of component levels are appropriate; however, these findings were not reported. In particular, the percentage of missing data on each item was not reported, yet this has implications in interpreting the findings.

The percentage of the missing data now included in the methodology section. The percentage of missing data relating to items ranged from (0.4 – 2.7 %).

Factor analysis may be appropriate, although a confirmatory factor analysis could have been more useful in validating the proposed factorial structure of the measure, which was derived from theory (health beliefs model).

The rationale for selecting: 1) principal component analysis instead of principal axis factoring (commonly used when factors represent concepts underlying responses to items), and 2) varimax rotation instead of oblique rotation (commonly used when the underlying factors are expected, in real world, to be correlated), is not presented. Please, explain the decisions.

Response: We added the rationale for selecting principal component analysis and varimax rotation in the manuscript.

The cut-off for eigenvalue is usually > 1.00 (not 0.3 as implied in the respective sentence) and for factor loading > 0.3 (though some use > 0.5). Please revise the sentence accordingly. Also, was a difference of 0.2 in the loadings of the same item on 2 factors used as a criterion to determine which factor the item loads on? Please, explain.

The statement was revised to reflect a correct eigenvalue greater than 1.

Regarding the specific item that cross-loaded with two factors, we retained the item in the factor in which it had a higher loading and removed the item from the other factor.

The Cronbach's alpha coefficient is set at > 0.70 for newly developed measure. What is the reason for setting it at 0.60 in this study?

Response: We have added the justification for selecting cut off points for the Cronbach's alpha.

The low (< 0.70) alpha coefficient value for the barriers subscale (or component) is attributable to the content of its items. The items assess different emotional, practical, service, and financial barriers. Participants may perceive these barriers differently. This also has implications for the interpretation of the findings and for the use of the measure in practice; in practice, it would be more meaningful to determine the specific type of barrier perceived by patients, as this inform the selection of most relevant intervention.

Response: We agree that barriers vary in nature and that a given barrier may have more, less or the same perceived salience for different respondents depending on a range of psychosocial health and life experiences. Arguably, given this point, it is surprising that a Cronbach's alpha for perceived barriers domain of 0.66 was achieved albeit less than the convention of < 0.7 . We interpreted 0.6 as indicating an acceptable moderate level of internal reliability (in keeping with other researchers eg Taber, 2018). We have included this thinking in the Discussion section of the paper including your important point about follow-on implications with which also we agree.

VERSION 2 – REVIEW

REVIEWER	De Guzman, Roselle Manila Central University-Filemon D Tanchoco Medical Foundation Hospital
REVIEW RETURNED	21-Mar-2023

GENERAL COMMENTS	Thank you to the authors for editing the manuscript accordingly and for taking into consideration the comments and suggestions of the reviewers. I appreciate the time it took them to revise to further improve the overall content and details of the manuscript.
---

REVIEWER	Gasper, Harry Toowoomba and Darling Downs Health Service District, Cancer Care
REVIEW RETURNED	27-Apr-2023

GENERAL COMMENTS	Many thanks for the opportunity to review the edited version of this manuscript. I have reviewed the response from the authors in tandem with the criticism from myself and each reviewer. The alterations and explanations made by the authors satisfy the requirements for publication and I would be happy to endorse the publication of the manuscript. I wish the best of luck to the authors with their study and look forward to seeing future results.
---

REVIEWER	Sidani, S Ryerson University
REVIEW RETURNED	24-Feb-2023

GENERAL COMMENTS	The revised manuscript addresses previous comments satisfactorily.
--